# Complex Posttraumatic Stress Disorder (CPTSD) as an Independent Diagnosis: Differences in Hedonic and Eudaimonic Well-Being between CPTSD and PTSD

**DOI:** 10.3390/healthcare11081188

**Published:** 2023-04-20

**Authors:** Danfeng Li, Jiaxian Luo, Xingru Yan, Yiming Liang

**Affiliations:** 1School of Sociology and Psychology, Central University of Finance and Economics, Beijing 100081, China; lidanfeng2018@126.com (D.L.); ljx311910@163.com (J.L.); yanxr2001@163.com (X.Y.); 2Shanghai Key Laboratory of Mental Health and Psychological Crisis Intervention, Affiliated Mental Health Center (ECNU), School of Psychology and Cognitive Science, East China Normal University, Shanghai 200062, China

**Keywords:** complex posttraumatic stress disorder, hedonic well-being, eudaimonic well-being, positive adaptation, posttraumatic growth

## Abstract

Although many studies have differentiated complex posttraumatic stress disorder (CPTSD) from posttraumatic stress disorder (PTSD), few studies have explored the differences in positive adaptation between the two. The present study aimed to determine whether there are distinctions between PTSD and CPTSD in hedonic and eudaimonic well-being. The present study used a Chinese young adult sample with childhood adversity experiences (*n* = 1451), including 508 males and 943 females, with an average age of 20.07 years (*SD* = 1.39). PTSD and CPTSD symptoms were measured by the International Trauma Questionnaire. Eudaimonic well-being was measured by the Meaning in Life Questionnaire, and hedonic well-being, including life satisfaction and happiness, was assessed by the Satisfaction with Life Scale and the face scale. Analysis of variance showed that the CPTSD group had lower hedonic and eudaimonic well-being than the PTSD group. Moreover, hierarchical regression analysis showed that disturbances in self-organization (DSO) symptoms in CPTSD were negatively associated with hedonic and eudaimonic well-being, while PTSD was positively associated with eudaimonic well-being. These findings indicate that the core symptoms of CPTSD might hinder individuals from living fulfilling lives. The positive association between eudaimonic well-being and PTSD symptoms may be a manifestation of posttraumatic growth. Based on the perspective of positive adaptation, these results provide new evidence of the importance of considering CPTSD as an independent diagnosis and suggest that future well-being interventions should be implemented in people with DSO symptoms.

## 1. Introduction

Complex posttraumatic stress disorder (CPTSD) is classified as a separate trauma-related disorder from posttraumatic stress disorder (PTSD) in the International Classification of Diseases (ICD-11) released by the World Health Organization in 2018 (WHO, 2018). Although CPTSD was not listed as an independent disorder until recently, it has long been considered distinct from PTSD [1]. Many studies have shown that PTSD is more common after exposure to one or more major catastrophic events, such as natural disasters, traffic accidents, or terrorist attacks [2]. In contrast, CPTSD often occurs after prolonged or repeated exposure to traumatic events from which escape is difficult or impossible (e.g., prolonged domestic violence; repeated sexual, physical, or emotional abuse; or captivity). For diagnosis with CPTSD, individuals must exhibit three symptom clusters of PTSD, namely (1) re-experiencing, (2) avoidance, and (3) a sense of ongoing threat. In addition, they must exhibit the three symptom clusters of disturbance in self-organization (DSO), namely (1) affective dysregulation, (2) negative self-concept, and (3) difficulties in relationships.

A recent systematic literature review showed that the construct validity of CPTSD includes two correlated second-order factors (PTSD and DSO) in clinical and community samples with a history of trauma exposure [3]. In addition, some scholars have used a personal-centered approach to investigate CPTSD symptom patterns, finding two main classes: CPTSD and PTSD [4,5,6]. Overall, these findings provide evidence that CPTSD is an independent disorder distinct from PTSD.

Previous studies have mainly focused on the negative impacts of CPTSD and PTSD on individuals, with CPTSD individuals exhibiting higher levels of psychopathological symptoms and risk behaviors than individuals with PTSD, including depression, anxiety, dissociation, sleep disturbances, somatization, interpersonal sensitivity, aggression, and substance abuse [6,7]. These findings suggest that PTSD and CPTSD differ in the severity of negative impacts on individuals.

Traumatic experiences may induce suffering, but positive adaptation may also occur after distressing events. Trauma-related distress may stimulate questioning and meaning-making [8]. To get their life back on track, individuals with traumatic experiences often make efforts to heal and overcome pain, which involve reappraisal of the traumatic event and seeking social support from an intimate relationship [9]. This process may help them find fulfillment in their lives, accomplish meaningful and worthwhile tasks, and connect with others at a deeper level, enabling them to flourish [10]. Notably, flourishing has been defined as a combination of high hedonic and eudaimonic well-being [11]. Previous research has found that well-being is a unique protective factor against future depression and anxiety [12,13]. Therefore, the differential impact of PTSD and CPTSD symptoms on well-being merits investigation. Such findings would enhance our understanding of the differences in positive adaptation in individuals with PTSD and CPTSD. However, research on this topic is scarce. One study explored the difference in positive adaptation between PTSD and CPTSD, focusing on prosocial behavior, but found no significant results [14]. Other positive adaptation aspects, such as hedonic and eudaimonic well-being, have not been evaluated.

Hedonic well-being refers to general satisfaction with present life [15], and it focuses on subjective cognitive-affective experiences of well-being [16]. Previous studies have found that PTSD symptoms are negatively correlated with hedonic well-being [17], and individuals with CPTSD have lower hedonic well-being than those with PTSD [18]. As a critical component of subjective well-being, life satisfaction is usually used to assess hedonic well-being [19]. Previous studies have found that PTSD symptoms and CPTSD symptoms are both negatively correlated with life satisfaction [20]. Moreover, the CPTSD group had a lower level of life satisfaction than the PTSD group and DSO group [21].

In contrast to hedonic well-being, eudaimonic well-being reflects self-realization, which involves subjective cognitive-affective experiences, such as the experience of meaning and purpose in life [22]. The core of eudaimonic well-being includes meaning in life, self-realization, and personal growth [23]. Meaning in life plays a critical role in individuals’ adaptation to traumatic life events [24]. Some studies conducted in military units have shown that meaning in life is negatively correlated with PTSD symptom severity [25,26]. However, these studies did not explore differences in meaning in life between PTSD and CPTSD. Prolonged exposure to traumatic events is an important characteristic of CPTSD and may exert a long-lasting influence on one’s daily life. However, PTSD is caused by sudden, short-term trauma. Thus, compared to PTSD, CPTSD may have a greater impact on eudaimonic well-being. The current study aimed to compare the differences in associations between eudaimonic well-being and CPTSD or PTSD symptoms.

The current study aimed to explore whether there are distinctions between ICD-11-diagnosed PTSD and CPTSD in hedonic and eudaimonic well-being in young adults. We compared hedonic and eudaimonic well-being between the PTSD and CPTSD groups. Moreover, we examined potential heterogeneity in the associations of PTSD and CPTSD symptoms with hedonic and eudaimonic well-being. In addition, we further separately examined the associations between well-being and CPTSD or PTSD symptoms in the CPTSD, PTSD, and DSO groups according to the diagnostic criteria of the ITQ. Based on previous studies, we hypothesized that (i) participants with CPTSD would have lower hedonic and eudaimonic scores than those with PTSD; (ii) CPTSD or PTSD symptoms would negatively predict hedonic and eudaimonic well-being; and (iii) compared with PTSD symptoms, CPTSD symptoms would predict lower levels of hedonic and eudaimonic well-being.

## 2. Methods

### 2.1. Participants and Procedure

The participants recruited in this study were college students. To obtain a representative sample of young adults, we employed a random stratified sampling procedure. The discipline type of universities was used to classify universities into 13 types. We also considered the running level of universities, attempting to include both key universities and average universities in each type as much as possible. Finally, we selected 29 out of 67 universities in Beijing. The distribution of the 29 included universities was as follows: Comprehensive (4), Science (5), Engineering (5), Agriculture (1), Normal (2), Finance and Economics (3), Forestry (1), Politics and Law (1), Medicine (1), Language (3), Nationality (1), Art (1), and Sports (1). The students were divided into strata in advance based on their university, major (liberal arts or sciences), and grades. The exclusion criteria for all participants were as follows: above the age of 27, presence of intellectual disability, a history of clinically significant head injury, or a history of neurological disorders such as encephalitis or epilepsy.

We contacted professors at the included colleges and requested their assistance in distributing questionnaires, until the number of completed questionnaires in each stratum reached the recruitment target. Each participant signed an informed consent form before accessing and completing the questionnaire online. The present study was conducted in accordance with the Declaration of Helsinki and approved by the Ethics Committee for Human Research of East China Normal University.

The current study used an online survey platform for data collection. A representative sample of 2048 participants from 29 universities completed the survey. The length of time needed to complete the entire questionnaire was approximately 15 min. A total of 224 participants were excluded because of careless completion (e.g., failure to pass the attention check items or providing the same answer to all items). The number of valid data points was 1824 (89.1%). We further excluded 373 participants who reported experiencing no traumatic experiences before 18 years of age according to the Life Events Checklist for DSM-5. The final sample in the current study comprised 1451 participants who had experienced childhood trauma. Participants were primarily female (64.9%, *n* = 942), and the mean age was 20.07 years (*SD* = 1.39). Other demographic variables are presented in Table 1.

### 2.2. Measures

#### 2.2.1. CPTSD

The International Trauma Questionnaire (ITQ) [27] was adopted to assess ICD-11-diagnosed PTSD and CPTSD. The Chinese version was revised by Ho et al. [28]. The ITQ consists of 18 items, twelve corresponding to the 12 symptoms of CPTSD and six items measuring functional impairment. PTSD symptoms (re-experiencing, avoidance, and sense of current threat) are assessed by six items, with each symptom measured by two items. Three items assess functional impairment associated with PTSD symptoms. Similarly, DSO symptoms (negative self-concept, affective dysregulation, and relationship disturbances) are evaluated by six symptom-related items and three function-related items. All items are rated on a five-point Likert scale ranging from 0 (not at all) to 4 (extremely). PTSD or DSO was diagnosed if each relevant symptom had a score of 2 (moderately) or greater. In addition, functional impairment was observed (at least one of the three items scored 2). CPTSD was diagnosed when both PTSD and DSO symptoms met the criteria. The Cronbach’s alpha for both PTSD and DSO symptoms in this study was 0.91.

#### 2.2.2. Childhood Trauma History

The Life Events Checklist for DSM-5 (LEC-5) was used to assess childhood trauma history [29]. The original scale presents 17 potentially traumatic events, such as natural disasters, physical or sexual assault, and serious injuries. Because some events in the original scale rarely appear in the Chinese social environment [30], we removed these four events (i.e., serious accidents at work, home, or during recreational activity; severe human suffering; combat or exposure to a war zone; and captivity). For each item, participants are asked to recall and indicate the type of exposure (e.g., whether they directly experienced or witnessed the event and whether it was related to occupational activities) before 18 years of age. Each item is rated on a six-point Likert scale, ranging from 0 (does not apply) to 5 (happened to me). Individuals who reported having witnessed or experienced at least one event were considered to have childhood traumatic experiences. The Cronbach’s α of this scale was 0.79 in this study.

#### 2.2.3. Eudaimonic Well-Being

Eudaimonic well-being was measured with the Meaning in Life Questionnaire (MLQ), which was developed by Steger et al. [31] and adapted for the Chinese context by Chan [32]. The MLQ contains two subscales, the presence of meaning and the search for meaning, with a total of 10 items. Participants are asked to respond to items on a seven-point scale ranging from 1 (strongly disagree) to 7 (strongly agree). One item, “My life has no clear purpose”, is scored in reverse; all other items are scored in a positive direction. The total score is the summation of all items and ranges from 0 to 70, with a higher score indicating a greater perception of meaning in life. In the present study, the scale exhibited good internal consistency (Cronbach’s α = 0.82).

#### 2.2.4. Hedonic Well-Being

The following two scales measured hedonic well-being. First, hedonic well-being was measured by the face scale developed by Andrews and Withey [33]. The scale is composed of seven faces, each with different expressions ranging from 1 (very sad) to 7 (very happy). The higher the score, the higher the level of subjective hedonic well-being. Second, hedonic well-being was also assessed using the five-item Satisfaction with Life Scale (SWLS) [34]. Answers were provided for the five items on a seven-point Likert scale, ranging from 1 (strongly disagree) to 7 (strongly agree). A higher value indicates a higher degree of life satisfaction. In this study, Cronbach’s α of this scale was 0.88.

### 2.3. Data Analysis

All data were analyzed in IBM SPSS (Version 23.0 for Windows). First, analysis of variance (ANOVA) was used to compare hedonic well-being (meaning in life) and eudaimonic well-being (life satisfaction and happiness) between the CPTSD group and the PTSD group. Second, correlation analysis and hierarchical regression analysis were conducted among participants with childhood trauma to explore the sources of differences between PTSD and CPTSD in hedonic and eudaimonic well-being. In hierarchical regression analysis, covariates (gender, age, father’s education level, and mother’s education level) were entered in Step 1, and PTSD symptoms and DSO symptoms of CPTSD were entered in Step 2. Finally, to further explore the different effects of PTSD symptoms and DSO symptoms on hedonic and eudaimonic well-being, we separately conducted hierarchical regression analysis (covariates entered in Step 1, and PTSD symptoms and DSO symptoms entered in Step 2) in the CPTSD, PTSD, and DSO groups according to the diagnostic criteria of the ITQ.

## 3. Results

### 3.1. Descriptive Statistics and Prevalence Rates of CPTSD, PTSD, and DSO Symptoms

A total of 1451 participants with childhood trauma were included in the final analysis. In selected participants who reported experiencing or witnessing at least one type of childhood trauma, based on the diagnostic criteria of the ITQ, the prevalence rates of CPTSD, PTSD, and DSO symptoms were 10.06% (*n* = 146), 6.00% (*n* = 87), and 9.51% (*n* = 138), respectively, in participants with childhood trauma. The mean number of traumatic events reported by participants was 3.41 (*SD* = 2.11), with 21.70% of participants reporting experiencing a single traumatic event and the majority (50.30%) reporting exposure to 2–6 traumatic events. The descriptive statistics of the main variables in the current study are presented in Table 2.

### 3.2. Differences in Meaning of Life, Life Satisfaction, and Happiness between the CPTSD Group and PTSD Group

Independent-sample t-tests were conducted to test whether there were significant differences between the CPTSD group and PTSD group in hedonic well-being (meaning in life) and eudaimonic well-being (life satisfaction and happiness) (see Table 3). Significant differences in meaning in life (*t* (231) = 4.63, *p* < 0.001), life satisfaction (*t* (231) = 2.42, *p* < 0.05), and happiness ((*t* (231) = 2.30, *p* < 0.05) between the two groups were observed. Specifically, participants in the PTSD group reported a higher level of meaning in life, life satisfaction, and happiness than participants in the CPTSD group.

### 3.3. Correlation Analysis and Hierarchical Regression Analysis among Participants with Childhood Trauma

Correlation coefficients were calculated between the main variables in the current study (i.e., meaning in life, life satisfaction, happiness, CPTSD symptoms, PTSD symptoms, and DSO symptoms). In addition to the total score of meaning in life, we also analyzed the scores of its two subscales, presence of meaning and search for meaning. The results of the correlation analyses of these main variables are presented in Table 2.

Three hierarchical regression models were used to examine how PTSD symptoms and DSO symptoms influenced participants’ hedonic well-being (life satisfaction and happiness) and eudaimonic well-being (meaning in life); see details in Table 4. In the first model for meaning in life, covariates (gender, age, father’s education level, and mother’s education level) explained 0.8% of the variance in meaning in life; *F* (4, 1446) = 2.81, *p* = 0.024, *R*^2^ = 0.01. PTSD symptoms and DSO symptoms explained 9.4% of the variance; *F* (6, 1444) = 27.15, *p* < 0.001, Δ*R*^2^ = 0.09. The coefficients in the final step indicated different effects of PTSD symptoms (β = 0.23, *p* < 0.001) and DSO symptoms (β = −0.36, *p* < 0.001) on meaning in life.

In the second model, covariates (gender, age, father’s education level, and mother’s education level) explained 2.4% of the variance in life satisfaction; *F* (4, 1446) = 8.82, *p* < 0.001, *R*^2^ = 0.02. Moreover, PTSD symptoms and DSO symptoms explained 19.9% of the variance; *F* (6, 1444) = 69.17, *p* < 0.001, Δ*R*^2^ = 0.20. The coefficients in the final step indicated different effects of PTSD symptoms (β = 0.11, *p* < 0.001) and DSO symptoms (β = −0.50, *p* < 0.001) on life satisfaction.

In the third model, covariates (gender, age, father’s education level, and mother’s education level) explained 0.4% of the variance in happiness; *F* (4, 1446) = 1.61, *p* = 0.169, *R*^2^ = 0.00. Furthermore, PTSD symptoms and DSO symptoms explained 25.5% of the variance; *F* (6, 1444) = 84.15, *p* < 0.001, Δ*R*^2^ = 0.26. The coefficients in the final step indicated different effects of PTSD symptoms (β = 0.03, *p* = 0.251) and DSO symptoms (β = −0.53, *p* < 0.001) on happiness.

### 3.4. Hierarchical Regression Analysis in the CPTSD, PTSD, and DSO Groups

Nine hierarchical regression models were used to explore the variance in eudaimonic well-being (meaning in life) and hedonic well-being (happiness and life satisfaction) in the CPTSD group, PTSD group, and DSO group; details are presented in Table 5 and Table 6. In the three hierarchical regression models for the CPTSD group, there was no evidence of a significant influence on meaning in life, *F* (6, 139) = 2.14, *p* = 0.052, Δ*R*^2^ = 0.05. Additionally, PTSD symptoms (β = 0.08, *p* = 0.364) did not significantly predict life satisfaction, whereas DSO symptoms (β = −0.33, *p* < 0.001) were found to be a negative predictor of life satisfaction; *F* (6, 139) = 5.33, *p* < 0.001, Δ*R*^2^ = 0.08. Moreover, PTSD symptoms (β = 0.02, *p* = 0.808) did not significantly predict happiness, whereas DSO symptoms (β = −0.29, *p* = 0.003) were found to be a negative predictor of happiness; *F* (6, 139) = 3.45, *p* = 0.003, Δ*R*^2^ = 0.07.

In the three hierarchical regression models for the PTSD group, PTSD symptoms (β = 0.30, *p* = 0.006) positively predicted meaning in life, whereas DSO symptoms (β = −0.30, *p* = 0.010) were found to be a negative predictor of meaning in life; *F* (6, 80) = 3.62, *p* = 0.003, Δ*R*^2^ = 0.12. Additionally, PTSD symptoms (β = 0.08, *p* = 0.389) did not significantly predict life satisfaction, whereas DSO symptoms (β = −0.54, *p* < 0.001) were found to be a negative predictor of life satisfaction; *F* (6, 80) = 6.85, *p* < 0.001, Δ*R*^2^ = 0.22. Moreover, PTSD symptoms (β = 0.01, *p* = 0.939) did not significantly predict happiness, whereas DSO symptoms (β = −0.49, *p* < 0.001) were found to be a negative predictor of happiness; *F* (6, 80) = 7.25, *p* < 0.001, Δ*R*^2^ = 0.19.

In the three hierarchical regression models for the DSO group, there was no evidence of a significant influence on meaning in life: *F* (6, 131) = 1.12, *p* = 0.352, Δ*R*^2^ = 0.01. Additionally, PTSD symptoms (β = −0.04, *p* = 0.659) did not significantly predict life satisfaction, whereas DSO symptoms (β = −0.23, *p* = 0.012) were found to be a negative predictor of life satisfaction; *F* (6, 131) = 2.31, *p* = 0.037, Δ*R*^2^ = 0.06. Moreover, PTSD symptoms (β = −0.11, *p* = 0.201) did not significantly predict happiness, whereas DSO symptoms (β = −0.29, *p* = 0.001) were found to be a negative predictor of happiness; *F* (6, 131) = 3.64, *p* = 0.002, Δ*R*^2^ = 0.10.

## 4. Discussion

Few studies have examined the independent diagnosis of CPTSD from the perspective of positive adaptation, such as well-being. The present study explored differences between CPTSD and PTSD in hedonic and eudaimonic well-being among young adults with childhood trauma. The main findings were as follows: (i) the CPTSD group had lower hedonic and eudaimonic well-being than the PTSD group, (ii) DSO symptoms in CPTSD had negative associations with hedonic and eudaimonic well-being, and (iii) PTSD had a positive association with eudaimonic well-being. Overall, the current study found that CPTSD was linked with worse hedonic and eudaimonic well-being than PTSD, and PTSD had a positive association with eudaimonic well-being.

As expected, the CPTSD group had lower hedonic well-being than the PTSD group. In particular, the CPTSD group had significantly lower happiness scores and marginally lower life satisfaction scores than the PTSD group. These findings related to hedonic well-being are consistent with previous studies [18,20]. Moreover, this study further found that the CPTSD group had significantly lower eudaimonic well-being scores than the PTSD group, reflected in both searching for meaning and the presence of meaning under meaning in life. Previous studies have mainly found that CPTSD has more severe negative psychological consequences than PTSD, such as depression, anxiety, sleep disturbances, and substance abuse [6,7]. The present study extends our understanding of the differences in positive adaptation of individuals between PTSD and CPTSD; CPTSD leads to greater disruption in hedonic and eudaimonic well-being. These findings emphasize the importance of considering CPTSD an independent diagnosis of PTSD in terms of positive adaptation.

To understand the differences in hedonic well-being between CPTSD and PTSD, we examined potential heterogeneity in the associations of DSO symptoms and PTSD symptoms with hedonic well-being. In the whole group, there was a negative association between DSO symptoms in CPTSD and life satisfaction, while PTSD symptoms and life satisfaction were positively associated. In the three groups meeting the ITQ diagnostic criteria, we further found a negative association between DSO symptoms in the CPTSD group and life satisfaction, while PTSD symptoms were not associated with life satisfaction. For happiness, we found similar results, both in the total group and in the three groups meeting the ITQ diagnostic criteria. DSO symptoms in CPTSD and happiness were negatively correlated, whereas PTSD symptoms were unrelated to happiness. Overall, these results show that DSO symptoms in CPTSD negatively predicted hedonic well-being, while PTSD symptoms had no significant correlation with hedonic well-being. These results may explain the difference in hedonic well-being between the CPTSD and PTSD groups—DSO symptoms may have a negative effect on hedonic well-being. DSO symptoms include affective dysregulation, negative self-concept, and difficulties in relationships. In terms of affective dysregulation, individuals with CPTSD may lack the ability to regulate emotion and feel less positive emotion, which may reduce their life satisfaction. Previous research has found that individuals with DSO symptoms report more emotional neglect than other groups [3]. Excessive emotional neglect may cause individuals to become depressed or anxious, decreasing their life satisfaction and happiness [35,36]. In terms of interpersonal aspects, patients with CPTSD may lack interpersonal support, which is considered a protective factor of mental health [37].

The current study further explored the differences between CPTSD and PTSD in eudaimonic well-being. We found that DSO symptoms in CPTSD were negatively related to meaning in life, while PTSD symptoms were positively associated with meaning in life in the whole group. In the three groups meeting the ITQ diagnostic criteria, we further found that PTSD symptoms positively predicted meaning in life, but DSO symptoms did not predict meaning in life. Overall, these findings show that PTSD symptoms positively predicted eudaimonic well-being. Moreover, compared with the general group that did not meet all diagnoses, the PTSD group showed a higher level of meaning in life (*t* (1514) = 3.53, *p* < 0.001). The higher level of eudaimonic well-being following PTSD symptoms may be a manifestation of posttraumatic growth, which has been defined as positive psychological changes following traumatic events [38]. These changes in mental functioning are often reflected in a greater sense of meaning in life and stronger social connections than before the traumatic event occurred [39]. Park’s integrated meaning-making mode [40] suggests that traumatic events may change individuals’ global meaning, which includes global beliefs, global goals, and subjective meaning in life. In addition, adverse experiences push individuals outside the realm of “normality” and provide them with multiple perspectives from which to see the world [41]. As a result, people with PTSD may spend more time thinking about the meaning of life. However, these results are inconsistent with a previous meta-analysis showing that PTSD symptoms were negatively correlated with eudaimonic well-being [26]. The following two aspects may explain this difference. First, compared with military personnel in previous studies [26], the sample in this study was mainly composed of university students with a higher level of education and cognitive ability; these characteristics might facilitate meaning construction. Second, the trauma experienced by military personnel is more severe than that of university students; indeed, war trauma is regarded as one of the most severe traumatic experiences, as people experience more danger to life and repeatedly witness death and destruction. These impacts on individuals are so severe that it destroys the system of meaning construction [42,43].

In contrast to PTSD symptoms, the results showed that DSO symptoms of CPTSD had a negative relationship with eudaimonic well-being, which provides new evidence of the distinction between PTSD and CPTSD in eudaimonic well-being. Compared with PTSD, CPTSD is more likely to result from long-term, inescapable trauma, such as domestic violence or repeated sexual or physical abuse, which leads individuals to believe that the world is unsafe and unmanageable. Terror management theory (TMT) holds that people need to generate and maintain the belief that the world is relatively predictable, orderly, and meaningful [44]. An unstable and inconsistent worldview may affect individuals’ understanding of the meaning of life. Thus, DSO symptoms may disrupt an individual’s recovery and subsequent growth [30].

The present study has several limitations. First, this study was cross-sectional, which prevents inference of a causal relationship between psychological symptoms and well-being. Longitudinal studies are needed to clarify this issue. Second, the current study focused on only young adults with childhood trauma. The differences in hedonic and eudaimonic well-being between CPTSD and PTSD need to be investigated in adults of other age groups. Third, although the ITQ is an effective tool for measuring CPTSD [27], the restriction characteristics of self-report questionnaires are still present. Structured interviews by clinicians should be used to provide more valid criteria for investigating symptoms. Fourth, we found a positive association between PTSD and eudaimonic well-being, possibly because traumatic events provide the possibility to change individuals’ concept of meaning in life. However, we did not explore the potential mechanisms underlying this relationship. We suggest that future studies explore potential factors underlying this relationship.

Despite these limitations, the present study made several contributions to theoretical research. First, this study found differences regarding hedonic and eudaimonic well-being between CPTSD and PTSD. Compared to PTSD, CPTSD is linked with lower hedonic and eudaimonic well-being, and these differences may be due to DSO symptoms. These findings indicate that CPTSD should be considered an independent diagnosis, as DSO symptoms may hamper positive adaptation among individuals suffering from CPTSD. Second, we found that PTSD had a positive association with eudaimonic well-being, possibly providing evidence of posttraumatic growth. This study also has practical implications. Considering the negative associations between DSO symptoms in CPTSD and well-being, clinical workers should precisely identify people with DSO symptoms and treat them with targeted interventions. For example, clinical workers can help these individuals rebuild their sense of autonomy and control, learn how to regulate emotions, and develop healthy relationships to reduce their DSO symptoms [21]. In addition, several effective well-being interventions are available for individuals with CPTSD, such as a life review program for hedonic well-being enhancement and reminiscence and photographic interventions to increase eudaimonic well-being [45,46,47].

## 5. Conclusions

The current study demonstrated differences in hedonic and eudaimonic well-being between CPTSD and PTSD. Compared to PTSD, CPTSD was linked with lower hedonic and eudaimonic well-being, which may indicate greater impairment of positive function. In addition, PTSD had a positive association with eudaimonic well-being, which indicates that traumatic events may provide an opportunity for changing individuals’ meaning in life.

## Figures and Tables

**Table 1 healthcare-11-01188-t001:** Participants’ demographic variables (*n* = 1451).

Characteristic	*n*	%	Characteristic	*n*	%
**Gender**			**Father’s education level**		
Male	509	35.1	Primary school	99	6.8
Female	942	64.9	Junior high school	230	15.9
**Grade**			High school	261	18.0
Freshman	398	27.4	Technical school	40	2.8
Sophomore	618	42.6	Secondary professional school	125	8.6
Junior	255	17.6	College	206	14.2
Senior	180	12.4	Undergraduate college	387	26.7
**Specialized subject**			Graduate student	91	6.3
Natural Science	290	20.0	Other	12	0.8
Agricultural Science	41	2.8	**Mother’s education level**		
Medical Science	61	4.2	Primary school	118	8.1
Engineering and Technical Science	310	21.4	Junior high school	263	18.1
Humanities and Social Sciences	749	51.6	High school	239	16.5
**Family residence**			Technical school	28	1.9
Provincial capital	614	42.3	Secondary professional school	196	13.5
City	275	19.0	College	229	15.8
County	333	22.9	Undergraduate college	306	21.1
Village	86	5.9	Graduate student	64	4.4
Hamlet	143	9.9	Other	8	0.6
**Family**					
Only child	899	62.0			
Non-only child	552	38.0			

**Table 2 healthcare-11-01188-t002:** Descriptive statistics and correlations among meaning in life, life satisfaction, happiness, and three symptoms (*n* = 1451).

	*M* (*SD*)	1	2	3	4	5	6	7	8
1. Meaning in life	51.02 (8.56)	1							
2. Search for meaning	27.69 (5.02)	0.69 ***	1						
3. Presence of meaning	23.32 (6.23)	0.81 ***	0.15 ***	1					
4. Life satisfaction	21.90 (6.68)	0.37 ***	0.05	0.47 ***	1				
5. Happiness	5.25 (1.17)	0.27 ***	0.06 *	0.33 ***	0.54 ***	1			
6. CPTSD symptoms	14.80 (10.48)	−0.09 ***	0.09 ***	−0.20 ***	−0.33 ***	−0.42 ***	1		
7. PTSD symptoms	7.32 (6.04)	0.06 *	0.11 ***	−0.01	−0.14 ***	−0.24 ***	0.88 ***	1	
8. DSO symptoms	7.48 (5.94)	−0.22 ***	0.04	−0.34 ***	−0.43 ***	−0.50 ***	0.87 ***	0.53 ***	1

Notes: CPTSD = complex posttraumatic stress disorder, PTSD = posttraumatic stress disorder, DSO = disturbances in self-organization; * *p* < 0.05, *** *p* < 0.001.

**Table 3 healthcare-11-01188-t003:** Differences in meaning in life, life satisfaction, and happiness between the CPTSD group (*n* = 146) and the PTSD group (*n* = 87).

	CPTSD Group (*n* = 146)	PTSD Group (*n* = 87)	*t*	*p*
Meaning in life	49.55 (7.63)	54.56 (8.58)	4.63 ***	<0.001
Search for meaning	27.23 (4.67)	28.85 (3.88)	2.73 **	0.007
Presence of meaning	22.33 (5.38)	25.71 (6.37)	4.33 ***	<0.001
Life satisfaction	20.33 (6.40)	22.54 (7.27)	2.42 *	0.016
Happiness	4.64 (1.53)	5.07 (1.04)	2.30 *	0.022

Notes: CPTSD = complex posttraumatic stress disorder, PTSD = posttraumatic stress disorder. Standard deviations are in parentheses. Means are outside parentheses. Different superscripts indicate statistically significant differences between various classes. * *p* < 0.05, * **p* < 0.01, *** *p* < 0.001.

**Table 4 healthcare-11-01188-t004:** Hierarchical linear regression analysis results for PTSD symptoms and DSO symptoms predicting hedonic and eudaimonic well-being (*n* = 1451).

*DV*: Eudaimonic Well-Being					*DV*: Hedonic Well-Being									
Meaning in Life					Life Satisfaction					Happiness				
Variables	Model 1		Model 2		Variables	Model 1		Model 2		Variables	Model 1		Model 2	
	β	*p*	β	*p*		β	*p*	β	*p*		β	*p*	β	*p*
Gender	−0.06 *	0.029	−0.05 *	0.041	Gender	−0.03	0.185	−0.03	0.140	Gender	0.06 *	0.022	0.06 *	0.012
Age	0.05	0.050	0.06 *	0.014	Age	−0.01	0.981	0.06 **	0.009	Age	−0.02	0.468	0.06 **	0.006
Father’s education	−0.02	0.615	−0.02	0.577	Father’s education	0.06	0.211	0.05	0.142	Father’s education	0.01	0.818	0.01	0.700
Mother’s education	0.04	0.240	0.04	0.249	Mother’s education	0.11 **	0.002	0.10 **	0.002	Mother’s education	−0.00	0.985	−0.01	0.673
PTSD symptoms			0.23 ***	<0.001	PTSD symptoms			0.11 ***	<0.001	PTSD symptoms			0.03	0.251
DSO symptoms			−0.36 ***	<0.001	DSO symptoms			−0.50 ***	<0.001	DSO symptoms			−0.53 ***	<0.001
*R* ^2^	0.01		0.10		*R* ^2^	0.02		0.22		*R* ^2^	0.00		0.26	
Δ*R*^2^	0.01		0.09		Δ*R*^2^	0.02		0.20		Δ*R*^2^	0.00		0.26	
*F*	2.81 *		27.15 ***		*F*	8.82 ***		69.17 ***		*F*	1.61		84.15 ***	

Notes: PTSD = posttraumatic stress disorder, DSO = disturbances in self-organization; * *p* < 0.05; ** *p* < 0.01, *** *p* < 0.001.

**Table 5 healthcare-11-01188-t005:** Hierarchical linear regression analysis results for PTSD symptoms and DSO symptoms predicting eudaimonic well-being.

Variables	*DV*: Meaning in Life											
	CPTSD Group (*n* = 146)				PTSD Group (*n* = 87)				DSO Group (*n* = 138)			
	Model 1		Model 2		Model 1		Model 2		Model 1		Model 2	
	β	*p*	β	*p*	β	*p*	β	*p*	β	*p*	β	*p*
Gender	−0.04	0.610	−0.00	0.978	−0.15	0.173	−0.08	0.454	−0.03	0.742	−0.03	0.707
Age	0.06	0.457	.07	0.411	0.21	0.053	0.15	0.155	0.14	0.113	0.11	0.208
Father’s education	−0.20	0.129	−0.19	0.137	0.25	0.102	0.29	0.055	−0.16	0.165	−0.17	0.146
Mother’s education	0.05	0.700	0.06	0.660	−0.20	0.195	−0.30 *	0.048	0.05	0.646	0.05	0.648
PTSD symptoms			0.24 *	0.016			0.30 **	0.006			0.09	0.310
DSO symptoms			−0.04	0.718			−0.30**	0.010			0.03	0.718
*R* ^2^	0.04		0.09		0.10		0.22		0.04		0.05	
Δ*R*^2^	0.04		0.05		0.10		0.12		0.04		0.01	
*F*	1.38		2.14		2.21		3.62 **		1.33		1.12	

Notes: CPTSD = complex posttraumatic stress disorder, PTSD = posttraumatic stress disorder, DSO = disturbances in self-organization; * *p* < 0.05; ***p* < 0.01.

**Table 6 healthcare-11-01188-t006:** Summary of hierarchical linear regression analysis for PTSD symptoms and DSO symptoms predicting hedonic well-being.

		CPTSD Group (*n* = 146)				PTSD Group (*n* = 87)				DSO Group (*n* = 138)			
		Model 1		Model 2		Model 1		Model 2		Model 1		Model 2	
		β	*p*	β	*p*	β	*p*	β	*p*	β	*p*	β	*p*
***DV*: Life satisfaction**	Gender	−0.12	0.159	−0.14	0.087	−0.04	0.700	0.02	0.817	−0.09	0.275	−0.08	0.343
	Age	0.22 **	0.009	0.23 **	0.005	−0.06	0.587	−0.16	0.107	0.01	0.894	0.05	0.537
	Father’s education	−0.19	0.131	−0.11	0.375	−0.14	0.375	0.04	0.753	0.24 *	0.039	0.24 *	0.040
	Mother’s education	0.24 *	0.049	0.18	0.130	0.43 **	0.006	0.20	0.156	−0.09	0.458	−0.05	0.691
	PTSD symptoms			0.08	0.364			0.08	0.389			−0.04	0.659
	DSO symptoms			−0.33 ***	<0.001			−0.54 ***	<0.001			−0.23 *	0.012
	*R* ^2^	0.11		0.19		0.12		0.34		0.04		0.10	
	Δ*R*^2^	0.11		0.08		0.12		0.22		0.04		0.06	
	*F*	4.15 **		5.33 ***		2.72 *		6.85 ***		1.46		2.31 *	
***DV*: Happiness**	Gender	0.11	0.185	0.08	0.318	0.18	0.089	0.23 *	0.018	−0.06	0.490	−0.04	0.622
	Age	0.11	0.220	0.11	0.184	0.21	0.052	0.12	0.211	0.09	0.311	0.15	0.070
	Father’s education	−0.16	0.208	−0.09	0.478	−0.35 *	0.021	−0.17	0.223	0.25 *	0.033	0.25 *	0.026
	Mother’s education	−0.04	0.756	−0.10	0.430	0.44 **	0.004	0.23	0.106	−0.21	0.069	−0.17	0.140
	PTSD symptoms			0.02	0.808			0.01	0.939			−0.11	0.201
	DSO symptoms			−0.29 **	0.003			−0.49 ***	<0.001			−0.29 **	0.001
	*R* ^2^	0.06		0.13		0.16		0.35		0.04		0.14	
	Δ*R*^2^	0.06		0.07		0.16		0.19		0.04		0.10	
	*F*	2.12		3.45 **		3.85 **		7.25 ***		1.41		3.64 **	

Notes: CPTSD = complex posttraumatic stress disorder, PTSD = posttraumatic stress disorder, DSO = disturbances in self-organization; * *p* < 0.05; ** *p* < 0.01, *** *p* < 0.001.

## Data Availability

The data presented in this study are available on request from the corresponding author.

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
