# Peer review of "Complex Posttraumatic Stress Disorder (CPTSD) as an Independent Diagnosis: Differences in Hedonic and Eudaimonic Well-Being between CPTSD and PTSD"

_healthcare, 2023, doi:10.3390/healthcare11081188_

Round 1

Reviewer 1 Report

The maina im of the study is to explore whether there are distinctions between posttraumatic stress disorder PTSD and CPTSD on hedonic and eudai- monic well-being.

Several qualitative aspects of the study can be highlighted.

- The research hypotheses are very clearly formulated.

- There is a theoretical model based on structural equations.

- The number of research participants is representative (2048 college students from 29 universities).

- The research instruments are applied for each defined variable - The International Trauma Questionnaire, The Life Events Checklist for DSM-5, the Meaning in Life Questionnaire, the 5-item Satisfaction with Life Scale.

- The statistical analysis of data aims the correlation analysis and hierarchical regression analysis among participants with childhood 242 trauma and hierarchical regression analysis in the CPTSD, PTSD, and DSO groups.

- The discussions are elaborated in close connection with the results of other current studies.

It is necessary to formulate conclusions and the implications of the research, as well as some limitations of the research. 

Author Response

Dear Reviewer 1,

Thank you very much for your invitation to revise our manuscript. Revised portions are marked in blue on the paper.

Several qualitative aspects of the study can be highlighted.

Comment 1: The research hypotheses are very clearly formulated.

Response: Thanks for your comments.

Comment 2: There is a theoretical model based on structural equations.

Response: Many thanks for your comments.

Comment 3: The number of research participants is representative (2048 college students from 29 universities).

 Response: We highlighted the representative sample of the present study in the Method.

 " A representative sample of 2048 participants from 29 universities completed the survey."

Comment 4: The research instruments are applied for each defined variable - The International Trauma Questionnaire, The Life Events Checklist for DSM-5, the Meaning in Life Questionnaire, and the 5-item Satisfaction with Life Scale.

 Response: In this revision, we clarified the measurement tools for each variable in the Abstract.

 " PTSD and CPTSD symptoms were measured by the International Trauma Questionnaire. Besides, eudaimonic well-being was measured by the Meaning in Life Questionnaire, and hedonic well-being, including life satisfaction and happiness, was accessed by the Satisfaction with Life Scale and the face scale."

Comment 5: The statistical analysis of data aims the correlation analysis and hierarchical regression analysis among participants with childhood trauma and hierarchical regression analysis in the CPTSD, PTSD, and DSO groups.

Response: Thanks for your comments. Correlation analysis and hierarchical regression analysis were conducted among participants with childhood trauma to explore the sources of differences between PTSD and CPTSD in hedonic and eudaimonic well-being. To further explore the different effects of PTSD symptoms and DSO symptoms on hedonic and eudaimonic well-being, we separately conducted hierarchical regression analysis in the CPTSD, PTSD, and DSO groups. We highlighted them in the Data analysis section.

" Data analysis: All data were analyzed in IBM SPSS (Version 23.0 for Windows). First, analysis of variance (ANOVA) was used to compare hedonic well-being (meaning in life) and eudaimonic well-being (life satisfaction and happiness) between the CPTSD group and the PTSD group. Second, correlation analysis and hierarchical regression analysis were conducted among participants with childhood trauma to explore the sources of differences between PTSD and CPTSD in hedonic and eudaimonic well-being. In hierarchical regression analysis, covariates (gender, age, father’s education level, and mother’s education level) were entered in Step 1, and PTSD symptoms and DSO symptoms of CPTSD were entered in Step 2. Finally, to further explore the different effects of PTSD symptoms and DSO symptoms on hedonic and eudaimonic well-being, we separately conducted hierarchical regression analysis (covariates entered in Step 1, and PTSD symptoms and DSO symptoms entered in Step 2) in the CPTSD, PTSD, and DSO groups according to the diagnostic criteria of the ITQ."

Comment 6: The discussions are elaborated in close connection with the results of other current studies. It is necessary to formulate conclusions and the implications of the research, as well as some limitations of the research. 

 Response: Many thanks for your comments. In this revision, we added a separate conclusion section, implications, and limitations.

 " Conclusion: The current study demonstrated differences in hedonic and eudaimonic well-being between CPTSD and PTSD. Compared to PTSD, CPTSD was linked with lower hedonic and eudaimonic well-being, which may indicate greater impairment of positive function. In addition, PTSD had a positive association with eudaimonic well-being, which indicates that traumatic events may provide an opportunity for changing individuals' meaning in life.".

Limitations: The present study has several limitations. First, this study was cross-sectional, which prevents inference of a causal relationship between psychological symptoms and well-being. Longitudinal studies are needed to clarify this issue. Second, the current study focused on only young adults with childhood trauma. The differences in hedonic and eudaimonic well-being between CPTSD and PTSD need to be investigated in adults of other age groups. Third, although the ITQ is an effective tool for measuring CPTSD [27], the restriction characteristics of self-report questionnaires were still present. Structured interviews by clinicians should be used to provide more valid criteria for investigating symptoms. Fourth, we found a positive association between PTSD and eudaimonic well-being, possibly because traumatic events provide the possibility to change individuals' meaning in life. However, we did not explore the potential mechanisms underlying this relationship. We suggest that future studies explore potential factors underlying this relationship.

Implications: Despite these limitations, the present study made several contributions to theoretical research. First, this study found differences regarding hedonic and eudaimonic well-being between CPTSD and PTSD. Compared to PTSD, CPTSD is linked with lower hedonic and eudaimonic well-being, and these differences may be due to DSO symptoms. These findings indicate that CPTSD should be considered an independent diagnosis, as DSO symptoms may hamper positive adaptation among individuals suffering from CPTSD. Second, we found that PTSD had a positive association with eudaimonic well-being, possibly providing evidence of posttraumatic growth. This study also has practical implications. Considering the negative associations between DSO symptoms in CPTSD and well-being, clinical workers should precisely identify people with DSO symptoms and treat them with targeted interventions. For example, clinical workers can help these individuals rebuild their sense of autonomy and control, learn how to regulate emotions, and develop healthy relationships to reduce their DSO symptoms [21]. In addition, several effective well-being interventions are available for individuals with CPTSD, such as a life review program for hedonic well-being enhancement and reminiscence and photographic interventions to increase eudaimonic well-being [45-47]."

Reviewer 2 Report

Dear editor,

Thank you so much for giving me the opportunity to contribute to the International Journal of Environmental Research and Public Research.

With less research on the complex posttraumatic stress disorder (CPTSD)the authors conducted an interesting study to explore whether there are distinctions between posttraumatic stress disorder PTSD and CPTSD on hedonic and eudaimonic well-being. We think this topic is very interesting and worth studying. I have just some minor comments.

1.Introductionwhen the DSO appears for the first time in the introduction section, it should be given its full name and written in brackets. The sentence "In terms of symptoms, CPTSD not only has the core symptoms of PTSD (re-experience, avoidance, and sense of threat) but also disturbances in self-organization” (DSO; affective dysregulation, negative self-concept, and difficulties in relationships)." is suggested to be slightly modified.

2.Introduction, the introduction section is overly complex and should be an overview of the background to the study rather than a list of previous findings. Streamlining the introduction section would make the content more readable and would allow the reader to learn more from it.

3.Methods, we recommend that the authors give more detailed exclusion criteria for the inclusion of participants.

4.Methods, the authors should give more details on ethical aspects of the paper (ethics committee review, the Declaration of Helsinki, privacy protection of participants, etc.)

5.Methods, please report the Cronbach's alpha of The Life Events Checklist for DSM-5 (LEC-5) in present study.

6.Result, table 5 has been suggested to change its format to a three-line table, which currently looks less aesthetically pleasing.

7.Discussion, some sentences are too long, for example, “The following two aspects may explain this difference: first, compared with military personnel in previous studies (e.g., Fischer et al., 2020), the sample in this study was mainly university students, and they had a higher level of education and cognitive ability, which facilitates their meaning construction; on the other hand, military personnel's trauma is more severe than that of university students, and war trauma is regarded as one of the most severe traumas, making people experience more danger to life and witnessing destructive and death scenes repeatedly.”, it is recommended to combine long and short sentences.

8.The author was advised to add a conclusion section to summarise the content of the text and give it a more complete structure.

9.Overall, I believe the manuscript may benefit from a thorough review of the language.

Author Response

Dear Reviewer 2,

Thank you very much for your invitation to revise our manuscript. Revised portions are marked in blue on the paper.

Comments and Suggestions for Authors

With less research on complex posttraumatic stress disorder (CPTSD), the authors conducted an interesting study to explore whether there are distinctions between posttraumatic stress disorder PTSD and CPTSD on hedonic and eudaimonic well-being. We think this topic is very interesting and worth studying. I have just some minor comments.

Response: Many thanks for your comments.

Comment 1: Introduction, when the DSO appears for the first time in the introduction section, it should be given its full name and written in brackets. The sentence "In terms of symptoms, CPTSD not only has the core symptoms of PTSD (re-experience, avoidance, and sense of threat) but also “disturbances in self-organization” (DSO; affective dysregulation, negative self-concept, and difficulties in relationships)." is suggested to be slightly modified.

Response: Thanks for your suggestion. We revised the sentence" "In terms of symptoms, CPTSD not only has the core symptoms of PTSD (re-experience, avoidance, and sense of threat) but also “disturbances in self-organization” (DSO; affective dysregulation, negative self-concept, and difficulties in relationships).", and gave DSO the full name when it appeared and wrote it in a bracket.

" For diagnosis with CPTSD, individuals must exhibit three symptom clusters of PTSD, namely, (1) re-experiencing, (2) avoidance, and (3) a sense of ongoing threat. In addition, they must exhibit the three symptom clusters of disturbance in self-organization (DSO), namely, (1) affective dysregulation, (2) negative self-concept, and (3) difficulties in relationships."

Comment 2: Introduction, the introduction section is overly complex and should be an overview of the background to the study rather than a list of previous findings. Streamlining the introduction section would make the content more readable and would allow the reader to learn more from it.

Response: We appreciate your comments and suggestions. In this revision, we streamlined the introduction section and removed the section listing studies.

Comment 3: Methods, we recommend that the authors give more detailed exclusion criteria for the inclusion of participants.

Response: Many thanks for your suggestions. We reported the exclusion criteria in detail:

“The exclusion criteria for all participants were as follows: above the age of 27, presence of intellectual disability, a history of clinically significant head injury, or a history of neurological disorders such as encephalitis or epilepsy.”

Comment 4: Methods, the authors should give more details on the ethical aspects of the paper (ethics committee review, the Declaration of Helsinki, privacy protection of participants, etc.)

Response: We appreciate your suggestion. We added the details on the ethical aspects of the paper:

“The present study was conducted in accordance with the Declaration of Helsinki and approved by the Ethics Committee for human research of East China Normal University.”

Comment 5: Methods, please report the Cronbach's alpha of The Life Events Checklist for DSM-5 (LEC-5) in the present study.

Response: Thanks for your comments. We added Cronbach's alpha of The Life Events Checklist for DSM-5 (LEC-5) in the revision.

" Cronbach's α of this scale was 0.79 in this study."

Comment 6: Result, table 5 has been suggested to change its format to a three-line table, which currently looks less aesthetically pleasing.

Response: Thank you very much for your comments. We changed table 5’s format to a three-line table in the revision.

Table 5. Hierarchical linear regression analysis results for PTSD symptoms and DSO symptoms predicting eudaimonic well-being.

Variables

DV: Meaning in life

CPTSD group (n = 146)

PTSD group (n = 87)

DSO group (n = 138)

Model 1

Model 2

Model 1

Model 2

Model 1

Model 2

β

p

β

p

β

p

β

p

β

p

β

p

Gender

−.04

.610

−.00

.978

−.15

.173

−.08

.454

−.03

.742

−.03

.707

Age

.06

.457

.07

.411

.21

.053

.15

.155

.14

.113

.11

.208

Father's education

−.20

.129

−.19

.137

.25

.102

.29

.055

−.16

.165

−.17

.146

Mother's education

.05

.700

.06

.660

−.20

.195

−.30*

.048

.05

.646

.05

.648

PTSD symptoms

.24*

.016

.30**

.006

.09

.310

DSO symptoms

−.04

.718

−.30**

.010

.03

.718

R2

.04

.09

.10

.21

.04

.05

R2

.04

.05

.10

.12

.04

.01

F

1.38

2.14

2.21

3.62**

1.33

1.12

Notes: CPTSD = complex posttraumatic stress disorder, PTSD = posttraumatic stress disorder, DSO = disturbances in self-organization; *p < .05; **p < .01, ***p < .001.

Comment 7: Discussion, some sentences are too long, for example, “The following two aspects may explain this difference: first, compared with military personnel in previous studies (e.g., Fischer et al., 2020), the sample in this study was mainly university students, and they had a higher level of education and cognitive ability, which facilitates their meaning construction; on the other hand, military personnel's trauma is more severe than that of university students, and war trauma is regarded as one of the most severe traumas, making people experience more danger to life and witnessing destructive and death scenes repeatedly.”, it is recommended to combine long and short sentences.

Response: We turned long sentences in Discussion into long and short sentences.

" The following two aspects may explain this difference. First, compared with military personnel in previous studies [26], the sample in this study was mainly composed of university students with a higher level of education and cognitive ability; these characteristics might facilitate meaning construction. Second, the trauma experienced by military personnel is more severe than that of university students; indeed, war trauma is regarded as one of the most severe traumatic experiences, as people experience more danger to life and repeatedly witness death and destruction. These impacts on individuals are so severe that it destroys the system of meaning construction [42,43]."

 Comment 8: The author was advised to add a conclusion section to summarize the content of the text and give it a more complete structure.

 Response: We added a separate conclusion section in this revision.

 " Conclusion

The current study demonstrated differences in hedonic and eudaimonic well-being between CPTSD and PTSD. Compared to PTSD, CPTSD was linked with lower hedonic and eudaimonic well-being, which may indicate greater impairment of positive function. In addition, PTSD had a positive association with eudaimonic well-being, which indicates that traumatic events may provide an opportunity for changing individuals' meaning in life.".

Comment 9: Overall, I believe the manuscript may benefit from a thorough review of the language.

Response: We used the American Journal Expert's (AJE) English retouching service to help us correct grammar, word choice, language, and punctuation, and further optimize writing style and fluency.

Round 2

Reviewer 2 Report

I do not have any other comments and consider that the article can be published.